# Prediction of The Mechanical Behavior of Polylactic Acid Parts with Shape Memory Effect Fabricated by FDM

**DOI:** 10.3390/polym15051162

**Published:** 2023-02-25

**Authors:** Zhamila Issabayeva, Igor Shishkovsky

**Affiliations:** Center for Materials Technologies, Skolkovo Institute of Science and Technology, 121205 Moscow, Russia

**Keywords:** shape memory polymer (SMP), fused deposition modeling (FDM), viscoelasticity, thermomechanical testing, biodegradable polymer, polylactic acid, 4D printing, mechanical strength

## Abstract

In this study, the mechanical as well as thermomechanical behaviors of shape memory PLA parts are presented. A total of 120 sets with five variable printing parameters were printed by the FDM method. The impact of the printing parameters on the tensile strength, viscoelastic performance, shape fixity, and recovery coefficients were studied. The results show that two printing parameters, the temperature of the extruder and the nozzle diameter, were more significant for the mechanical properties. The values of tensile strength varied from 32 MPa to 50 MPa. The use of a suitable Mooney–Rivlin model to describe the hyperelastic behavior of the material allowed us to gain a good fit for the experimental and simulation curves. For the first time, using this material and method of 3D printing, the thermomechanical analysis (TMA) allowed us to evaluate the thermal deformation of the sample and obtain values of the coefficient of thermal expansion (CTE) at different temperatures, directions, and running curves from 71.37 ppm/K to 276.53 ppm/K. Dynamic mechanical analysis (DMA) showed a similar characteristic of curves and similar values with a deviation of 1–2% despite different printing parameters. The glass transition temperature for all samples with different measurement curves ranged from 63–69 °C. A material crystallinity of 2.2%, considered by differential scanning calorimetry (DSC), confirmed its amorphous nature. From the SMP cycle test, we observed that the stronger the sample, the lower the fatigue from cycle to cycle observed when restoring the initial shape after deformation, while the fixation of the shape did not almost decrease with each SMP cycle and was close to 100%. Comprehensive study demonstrated a complex operational relationship between determined mechanical and thermomechanical properties, combining the characteristics of a thermoplastic material with the shape memory effect and FDM printing parameters.

## 1. Introduction

The widespread use of additive manufacturing (AM) techniques has been linked to the fact that AM technologies save product development costs and time. The mechanical performance of the most widely used polymers is often the factor limiting possible applications because the AM process and parameters influence the resulting mechanical properties, which could be radically different from the unprocessed material. The most common additive manufacturing technique is fused deposition modeling (FDM), where a material is layered from a molten thermoplastic filament extruded by a heated nozzle. By adjusting several variables of the printing regime including the infill, density, filament orientation, and print speed, the FDM technique allows for the mechanical qualities to be customized. According to the literature, the parameters, which include printer settings and part characteristics, have a direct impact on the strength properties of the part [1,2]. Customizing printing parameters is a delicate procedure with numerous aspects that influence the quality and characteristics of the components. Printing and process parameters influence the bonding process between adjacent filaments that, in turn, influence the void density and bond strength, which means directly defining the mechanical properties of the FDM components. Some parameters have a negligible effect, while others are decisive. For example, the height of the layer affects the interlayer bonding, and the nozzle temperature affects the detail and strength. Furthermore, the rougher the surface of the previous layer of the printout, the better its adhesion with the subsequent layer will be. Better adhesion can be achieved by adjusting the bed temperature and even the print speed, as examined by Vannamei, El Magri et al. [3]. They noticed that the variation of 40 °C in the platform temperature resulted in an 11% increase in tensile strength and the increase of 20 mm/s in print speed resulted in a 14% increase.

Rankouhi et al. [4] examined the effect of the thickness and orientation of the layer on the tensile properties. It was found that the acrylonitrile butadiene styrene (ABS) part with a layer thickness of 0.2 mm showed higher tensile strength properties than with a layer thickness of 0.4 mm. A similar study presented by Wang [5] concluded that a smaller layer thickness increased the tensile strength due to smaller interlayer gaps and fewer air pores in the cross-section. Chacon et al. [6], in contrast, argued that in the vertical samples, the tensile and bending strength increased as the layer thickness increased. A more reliable approach would be to estimate the ratio of the nozzle diameter and the layer height and its effect on the strength of the part. This ratio is characterized by the area of the interlayer contact surface [7]. An increase in the ratio of the nozzle diameter to the height of the layer results in a cross-section of the thread similar to a circle; with a decrease in the layer’s height, the cross-section of the thread resembles a rectangle. Most of the air remains at the junction of the perimeters, which reduces the strength of the printout.

Regarding the filling pattern, the published study by Khan et al. [8] showed greater strength for ABS parts with a rectilinear infill pattern, while a concentric infill pattern yielded the best elongation, and the honeycomb pattern was recognized as irrational due to a waste of material. Yeoh et al. showed that different filling patterns influenced the strength and flexibility of the part to varying degrees [9], and PLA with a zigzag infill pattern showed the highest tensile strength in comparison with grid and concentric patterns.

The 3D printing of smart materials such as shape memory polymers (SMP) has led to the development of 4D printing technology [10]. The novelty of the topic is that the shape of 3D printed products can be transformed over time, achieving 4D printing. The difference is that the 3D object is static, but the 4D object is already dynamic. In this case, the fourth dimension is the response time to the stimulus and the subsequent change in shape [11]. Shape memory materials have the capability to change their shape upon the application of an external stimulus [12]. The ability to physically change the size and shape, particularly compression, expansion, and twisting, has allowed for the use of SMPs for artificial muscles, parts for aerospace ships, transforming medical devices, and intelligent textiles. In our recent review [13], it was shown that the 4D printing of SMP opens up the bio-fabrication possibilities of hierarchical auxetic metamaterials. There are artificial devices such as coronary stents, filtration and drug delivery systems, implants and many others where the ability for high-precision tuning are highlighted.

Typically, the materials used for 4D printing have a SME. However, a group of scientists has proposed a new approach to 4D printing without SMP and extra operations such as synthesis and blending [14,15]. They created encapsulated and bilayer macroscopic SMP structures (PCL-TU) using multimaterial printing of nonshape memory polymers. This solved the problem of a limited selection of printable SMP materials, making it possible to use all commercial filaments.

The SME in a material can be estimated by the recovery coefficient from cycle to cycle. Therefore, it is appropriate to emphasize that FDM parameters can impact the shape recoverability. In [16], Liu et al. showed that the thickness of the sample layer printed on a 3D printer had a significant impact on the recovery efficiency. The lower the layer height, the higher the elastic modulus and yield strength corresponding to it. The choice of nozzle temperature also plays an important role. Therefore, it was revealed that lower printing temperatures and higher printing speeds improve the characteristics of self-bending and shape recovery [17,18]. In another study, the authors summarized that the combination of a lower nozzle temperature with a higher printing speed had a negative effect on the recovery time and percentage, since high-viscosity materials have a limited time for extrusion as a result of the fast printing process [19]. Soleyman et al. [20] investigated the 4D printing capability of shape memory thermoplastic PET as well as the influence of printing temperature and speed on the emergent curved third shape during recovery. Despite the different values of the printing parameters and the degree of the polymer chain extension, the shape recovery for the two samples exceeded 96%.

As above-mentioned, the ability to keep and restore shape characterizes SMPs, which is due to the glass transition of thermally activated amorphous SMPs. The amorphous shape memory effect (SME) is dependent on molecular interactions [21]. Structurally, polymers look like long flexible macromolecules capable of changing the spatial arrangement of atoms under the influence of mechanical and thermal action, while the mobility and speed of conformations depend on the temperature regime [22]. Temperature has a considerable influence on the mechanical characteristics of polymeric materials [23]. At higher temperatures, an amorphous polymer that is stiff at ambient temperature becomes bendable and stretchy. The mechanical characteristics of the SMPs change with temperature, particularly around their glass transition temperatures. The glass transition temperature is the temperature at which the polymer chains become more mobile and there is a transition from a glassy rigid state to a rubbery deformable state [24]. The glass transition temperature is an essential result of dynamic mechanical analysis (DMA) as well as differential scanning calorimetry (DSC) and thermomechanical analysis (TMA). In fact, the value is not strictly defined, as depending on the way that it is determined, the value will vary [25]. There are three ways to determine the glass transition temperature from DMA data, and all three curves can show their own value of a given temperature. The T_g_ can be determined by the drop in the storage module, the peak of the loss module, or the peak of the tan(δ). Viscoelastic properties are manifested at temperatures exceeding T_g_.

The mechanical properties of specimens from the same PLA material can vary significantly. In our study for the above-mentioned material, we not only obtained the mechanical characteristics for different printing parameters, but also investigated the thermal behavior of the FDM printed samples providing TMA + DSC as well as DMA analyses. Aside from everything else, we conducted cyclic thermomechanical tests to study the SME. The results were brought together to create a more detailed portrait of the PLA material, its parts, and its behavior under different conditions. Furthermore, we were able to compare the experimental results of the tensile test and DMA with the FEM calculations.

## 2. Materials and Methods

### 2.1. Materials

In all stages of the research, the material used was a SMP from polylactic acid—PLA (Createbo, Ningbo, China). Despite the variety of printing materials, PLA is the most usable and widespread as a building material in FDM [26]. PLA is a synthetic aliphatic polyester derived from agricultural resources. PLA is favored due to its biodegradability, renewability, and its low toxicity, so is strongly recommended for medical applications. The material has a lower shrinkage and melting temperature than other 3D printing polymers, making polylactic filament easier to extrude through the FDM nozzle. The PLA filament is an amorphous polymer in which the glass transition temperature is a critical point of phase transition, and hence the critical temperature to study the SME.

There are three types of the polylactic acid structure due to the different optical rotation, L-polylactic acid, DL-polylactic acid, and D-polylactic acid [27]. The degree of amorphous branching inside the polymer chains differs. Both PLLA and PDLLA polymers are naturally crystalline with a glass transition temperature of 60 °C, which means that they take on ordered molecular structures. Poly(DL-Lactide) PDLLA is an amorphous polymer with a glass transition temperature of about 55 °C. According to the supplier, the type of material used in this project was PDLLA.

### 2.2. Uniaxial Tensile Tests

#### 2.2.1. Sample Fabrication

A dog-bone shape was chosen as the specimen geometry with dimensions of the rectangular part of 60 mm in length, 3 mm in width, and 3 mm in thickness according to ASTM D638-10. Five printing process parameters were selected. For each parameter, the following values were defined based on the values reported in the literature and the capabilities of the FDM systems used in this research:Infill density: 20%, 25%, 50%, 70% and 80%;Infill pattern: grid, line, gyroid;Layer thickness: 0.2 mm, 0.3 mm;Diameter of the nozzle: 0.3 mm, 0.5 mm;Temperature: 210 °C, 240 °C.

Other printer parameters remained constant throughout the study. The bed temperature was considered to be equal to 60 °C. The printing speed was set to 60 mm/s. Printing parameters were chosen based on the preliminary literary analysis and our own experience in working with a 3D printer. The literature states that the most important printing parameters are layer thickness, infill, and temperature [28,29]. The values for the infill density were selected based on preliminary attempts to evaluate the quality of the printed parts. Therefore, details with a density of less than 20% were unacceptable, and a huge difference in strength was not observed between 80% and 100%, where only time consumption and waste material were highlighted.

According to the parameters and values selected, 120 combinations were fabricated using a Picasso Designer X Pro 3D printer (see Appendix A). Every regime comprised 4 of the same specimens, which means that 480 specimens were printed and evaluated for the tensile test.

#### 2.2.2. Uniaxial Tensile Experiments

The uniaxial tensile tests were performed following the ASTM D638-02 standard. Uniaxial tensile tests were performed on an Instron 5969 universal material testing machine (Instron, Norwood, MA, USA). Displacements of the center section were measured with an extensometer with a gauge length of 25 mm. The allowable travel of the extensometer was 2.5 mm, which corresponded to the limit of measuring strain of 10%. The basic experimental conditions were a room temperature of 22 ± 0.5 °C, and a relative humidity of around 35%.

The initial sample dimensions (width and thickness) were measured. The specimens were settled into the grips of the testing machine. A load at a constant speed of 1 mm/min was applied. A gradual load of 1.0 kN was applied to the samples until the moment of failure, and the resulting loads and displacement were recorded by the testing machine for each individual sample. The tests ended automatically when the specimens fractured.

### 2.3. Differential Scanning Calorimetry

Differential scanning calorimetry (DSC) is the most commonly used method of thermal analysis, in which the difference in the amount of heat required to increase the temperature of the sample and the reference is measured as a function of temperature. The value of measuring the energy flow is that it allows us to determine the range of different transitions that can occur in a sample when it is heated or cooled. To avoid temperature gradients inside the sample, a small sample size is preferable. A DSC test was conducted on the ASTM D3418 standard. The dimensions of the studied sample were 3 mm × 3 mm × 3 mm. A sample was heated or cooled and the changes in its heat capacity were tracked as changes in the heat flow. This allows for the detection of transitions such as melts, glass transitions, phase changes, and curing. The experiment was carried out on the DSC3+ Excellence system (Mettler Toledo, Greifensee, Switzerland). To conduct the measurements, the sample was placed in a calorimetric crucible and sealed. The DSC temperature program was determined in the range of 0–180 °C with a heating rate of 10 K/min in a nitrogen medium with a flow rate of 50 mL/min in an Al crucible at 40 µL.

### 2.4. Thermomechanical Analysis

The TMA experiment was carried out on the TMA/SDTA2+ system (Mettler Toledo, Greifensee, Switzerland) under static load using various sensors to measure the changes in the length of the sample depending on temperature. The most important measurements of TMA include the determination of the coefficient of linear thermal expansion—α and the glass transition temperature T_g_ (ASTM E831 and ASTM E1545). The size of the test sample experiment should not exceed 4–5 mm. The dimensions of the tested sample were 3 mm × 3 mm × 3 mm. It is important to note that the initial dog-bone specimen was cut in the central part into cube form. The surface was pre-polished to achieve smoothness. The experiment program was based on the results of the DSC measurements and consists of several consecutive segments:Isotherm 0 °C during 10 min;First heating 0–90 °C;Cooling 90–0 °C;Second heating 0–150 °C.

The heating rate was 3 K/min in a nitrogen medium with a flow rate of 30 mL/min, and a probing constant load of 0.02 N in the dilatometric mode. Purge gas, N_2_, provided a continuous laminar gas flow to prevent the formation of air turbulence when the temperature rose, prevent the deposition of decomposition products inside various parts of the device, increase heat transfer to the sample, and prevent oxidation at high temperatures. To measure the internal properties of a material, the thermal history of the samples must first be erased. The treatment was performed in a thermal analysis device by heating the sample to a temperature exceeding its glass transition temperature.

### 2.5. Dynamic Mechanical Analysis

The DMA tests were conducted using a DMA Q800 machine (TA Instruments, Houston, TX, USA) to determine the glass transition temperature T_g_ of the SMP specimen as described in ASTM D4065. Four specimens with different printing configurations were selected for analysis. A rectangular sample with dimensions of 60 mm × 5 mm × 2 mm was placed in the DMA device using a dual cantilever clamp configuration. A small dynamic load with an amplitude of 10 μm with a frequency of 1 Hz was applied to the roller. The sample was examined in the range from room temperature to 150 °C. The heating rate of 3 K/min was used to avoid the effects of a fast temperature rise on the experimental results. The data were collected every 3 s during thermomechanical analysis.

### 2.6. Thermomechanical Shape Memory Cycle

Uniaxial tensile tests were carried out on a servo-hydraulic testing machine Instron 8801 (Instron, Norwood, MA, USA) equipped with a temperature chamber and an optical window to conduct observations inside the chamber. The overall length was set to 210 mm, the gauge length of sample was 60 mm, and the cross-section was 50 mm × 7 mm. Deformation control was carried out using the Digital Image Correlation method (DIC) [30,31]. Vic-3D (Correlated Solutions, Irmo, SC, USA) software was used to keep track of the shooting and process the photos that were captured. Figure 1 shows sample no. 17 during the first and fifth cycles with a field of measurement in the DIC system before and after mechanical testing.

A typical cycle to study the SME is a sequence of heating–loading–cooling–unloading–heating steps [32]. In this study, this heating–cooling process was carried out on up to five shape memory cycles. A specimen was heated up to 70 °C with rate of 3 °C/min and subsequently deformed with 60% strain, then allowed to cool down, keeping the deformation. The cooling rate was set to 3 °C/min. The length of the sample during stretching was set to 60 mm, and the stretching speed was 50 mm/min.

After heating, the recovery ratio could be calculated from the recovered shape of the samples. Tandon et al. proposed the following definition of the shape memory ratios [24]:(1)$$R_{fix}=\frac{\varepsilon_{u}}{\varepsilon_{p}}\cdot 100\%,$$
where $R_{fix}$ measures the ability of the specimens to fix the shape; $\varepsilon_{u}$ is a strain obtained after unloading; and $\varepsilon_{p}$ represents the maximum strain. The shape recovery ratio is expressed as:(2)$$R_{rec}=\left(1-\varepsilon_{f}\right)\cdot 100\%,$$
where $\varepsilon_{f}$ is the final strain expressed through the relation between the change in length after recovery and the initial specimen length.

### 2.7. Numerical Simulations

#### 2.7.1. Uniaxial Tensile Test

The finite element method to solve basic control equations and related boundary conditions specific to a particular task was implemented in Ansys Workbench 2019 (Ansys Inc, Canonsburg, PA, USA). Finite element analysis (FEA) was used to simulate the tensile load on the sample up to the moment before destruction. The built-in library contains a large selection of various materials including PLA. However, for nonlinear analysis, only having embedded data is not sufficient. To process and apply experimental data as input, it is necessary to define a model. Here, we considered the Mooney–Rivlin model with 3, 5, and 9 parameters [33]. The model with 3 parameters showed the worst fit, while 5 and 9 parameters fitted the curve equally well. Therefore, we chose the Mooney–Rivlin model with 5 parameters as the determining one. The coefficients of the five-parameter Mooney–Rivlin model were found by approximating the experimental data using Ansys Workbench. The specimens were meshed at the resolution of 5 mm/cell with HEX20 elements. The load was applied to the upper plane of the sample and the boundary conditions were applied to the lower part.

#### 2.7.2. Viscoelastic Behavior

PLA material showed viscoelastic behavior, as studied by DMA. This viscoelastic behavior is typical for amorphous polymers at temperatures close to the glass transition. Linear models are most often used to describe viscoelastic behavior. The linear viscoelastic Prony model is relatively simple, and at the same time, quite informative [34]. The Prony model contains a time relaxation series for each component, so relaxation occurs not at once, but over a certain time. This model, or rather the components characterizing it, can be specified in the finite element calculations of structures using the Ansys Workbench software. Based on the limited amount of data obtained experimentally, it is possible to determine the rheological parameters of the viscoelastic model.

Here, we considered the method to calculate the Prony series from DMA data. The determining relation is represented by the viscoelastic Prony model in the case of the absence of volumetric relaxation [35]:(3)$$G\left(t\right)=G_{0}\left[\alpha_{0}^{G}+\sum_{i=1}^{N}\alpha_{i}^{G}\exp\left(-\frac{t}{\tau_{i}^{G}}\right)\right],$$
where G(t) is the shear relaxation modulus; $\alpha_{i}^{G}$ are the relative shear modulus for the shear relaxation times $\tau_{i}^{G}$; N is the number of shear relaxation times.

In the equation described above, it is assumed that the material only experiences shear relaxation and is characterized by a constant volume compression modulus. Often, to obtain experimental parameters, test data are used not for shear, but for uniaxial tension, as was also used in our study. Therefore, the relaxation module for uniaxial stretching will be represented as [35]:(4)$$E\left(t\right)=E_{0}\left[c_{0}+\sum_{i=1}^{N}c_{i}\exp\left(-\frac{t}{\beta_{i}}\right)\right],$$
where $c_{i}$ is the relative module of tension–compression for relaxation times $\beta_{i}$; N is the number of the shear relaxation times of tension–compression relaxation; $E_{0}$ is a modulus when t=0. All $c_{i}$ must be positive and in sum <1 [36].

Using the calculation part with related equations in the approach by Smetannikov et al., we assumed [35]:(5)$$E^{\prime }=E_{\infty }+\omega^{2}E_{0}\sum_{i=1}^{N}\frac{{\beta^{\prime }}_{i}^{2}c_{i}}{1+{\beta^{\prime }}_{i}^{2}\omega^{2}},$$
(6)$$E^{\prime \prime }=\omega E_{0}\sum_{i=1}^{N}\frac{\beta_{i}^{\prime }c_{i}}{1+{\beta^{\prime }}_{i}^{2}\omega^{2}},$$
where $E^{\prime }$ and $E^{\prime \prime }$ are the real and imaginary parts of the complex modulus of elasticity; $E_{\infty }$ is a modulus when t=∞. The relaxation times given are in accordance with the principle of the temperature–time analogy:(7)$$\beta_{i}^{\prime }=\frac{\beta_{i}}{A\left(t\right)},$$
where A(t) is the shift function. Time–temperature superposition is achieved by the William–Landel–Ferry (WLF) shift factor as given in the next equation [35,36]:(8)$$\mathrm{lg}\left(A\left(T\right)\right)=\frac{C_{1}\left(T-T_{r}\right)}{C_{2}+\left(T-T_{r}\right)},$$
where lg(A(T)) is the shift factor in logarithmic form; $T_{r}$ represents the glass transition temperature obtained from the experiment; the constants $C_{1}$ and $C_{2}$ are empirical constants of the material and approximated as 17.44 and 51.6, respectively [36].

The storage and damping properties of the generalized Maxwell model (GMM) [36] were used to estimate the master curve for viscoelastic behavior. The GMM model with a number of Maxwell elements using Prony series coefficients of relaxation time $\beta_{i}$ and weights $\alpha_{i}^{G}$ fit the experimental data. For practical use, curve fitting can be implemented using software, however, it is necessary to determine the optimal number of Prony components. Each commercial CAM software package may have a different definition of the Prony series. We used the Origin 2021 software (Origin Lab, Northampton, MA, USA) to set the user-defined function, which contained a viscoelastic equation (Equation (5)) for the storage modulus, to fit a nonlinear curve.

#### 2.7.3. Finite Element Modeling of DMA

The mechanical properties of the proposed model, which depend on temperature, were evaluated via step-by-step modeling of the DMA experiment with a dual cantilever. Along with determining the Prony coefficients, the thermomechanical characteristics should also be considered. The PLA material specification in different temperatures were into engineering data of Ansys. The defining relations corresponded to the viscoelastic Prony model using the Williams–Landell–Ferry shift function (Equation (8)) in the calculation.

The preliminary convergence study with refined meshing allowed the appropriate mesh size to be set to 1 mm. The mesh was created using TET10 elements. During the simulation, a rectangular sample was subjected to cyclic loading in accordance with a sinusoidal waveform at a frequency of 1 Hz with an amplitude of 10 microns. The displacement of 10 × 10^−3^ mm was applied to the small cross-section of the specimen. Boundary conditions were applied both to the upper surface and to the lower one. The value of the dynamic elastic modulus was calculated as [37]:(9)$$E_{dyn}\left(\omega \right)=\left|E^{*}\left(\omega \right)\right|=\frac{\sigma_{0}}{\varepsilon_{0}\left(\omega \right)},$$
where $\sigma_{0}$ and $\varepsilon_{0}$ are the stress and strain amplitude, respectively.

## 3. Results

### 3.1. Thermomechanical Analysis and Differential Scanning Calorimetry

The thermogram of the PLA sample contained several thermal events, as shown in Figure 2.

The printed material was examined in three directions relative to the original shape of the bar:Normal to the sample is the thickness (Z coordinate of the 3D printer);Across the sample is the width (X coordinate of the 3D printer);Along the sample is the length (Y coordinate of the 3D printer).

The coefficient of thermal expansion (CTE) was determined from the TMA measurements. From the upper part of Figure 2, we observed how significant different CTE was not only found in different directions, but also in the first heating, second heating, and cooling run. The cyclical heating–cooling–heating method is commonly applied in TMA studies to eliminate the thermal history and reduce the internal stresses in the sample. The transition temperatures are determined by the sample’s internal stresses, and hence its thermal history. This explains why the curves observed in the first and second heating runs had distinct forms.

As it turned out, the most dramatic effects were observed in the first heating, while geometry changes not related to thermal expansion began to take place already at temperatures higher than 41 °C. The nature of the change in the measured deformation effect differed significantly for the three directions of the coordinate grid of the sample, which indicated a serious CTE anisotropy of the studied material. Without considering the thermal expansion, the thickness of the sample (Z coordinate black) decreased (maximum) by 1.3%, the width (X coordinate red) decreased by 0.07%, and the length (Y coordinate blue) increased by 0.7%. Furthermore, the sample sought to compensate for the deformation effect, which may be associated with the beginning of the crystallization process.

The cooling and second heating curves looked more predictable. While cooling, the CTE changed in three directions and was less visible: the normal direction at a glassy state had a CTE equal to 75.97 ppm/K and above the T_g_, the CTE was equal to 269.93 ppm/K and 251.72 ppm/K for the normal and across directions, respectively. The second heating run curve was examined in the same way. In the glassy state, the CTE was 74.36 ppm/K for the normal direction, 69.9 ppm/K for the across direction, and 71.37 ppm/K for the along direction. Above the glass transition temperature, the CTE showed 276.53 ppm/K for the normal direction, and 205.8 ppm/K for both the along and across directions. The CTE values are necessary in the process of modeling the behavior of the material in order to take into account the change in stress with temperature changes. From [38], the value of CTE is 98 ppm/K in the glassy state. This is more than the value we received of 14–20 ppm/K. The changes in the CTE values are shown in Table 1.

The glass transition temperature itself was well fixed on the cooling (54.55 °C) and heating (54.22 °C) curves and located as the intersection of tangents corresponding to the change in the CTE before and after the phase transition. Residual anomalies of geometry change continued to be observed in the second heating curve in the range of 85–105 °C. After 147 °C, the sample melted and began to deform under the influence of probing stress.

The DSC curve is shown at the bottom of Figure 2. In the range of 55–60 °C, a jump in the heat capacity (Delta C_p_) of 0.57 J/(g K) was observed, corresponding to the glass transition of the polymer with onset T_g_ = 55.9 °C, midpoint T_g_ = 58.2 °C, and endpoint T_g_ = 60.5 °C. Then, in the range of 93–135 °C, there was an exothermic peak of polymer crystallization at 115.15 °C, followed by the latent heat of crystallization of about 23.97 J/g. After 135 °C, melting occurred with an endothermic peak at 151.4 °C and the latent heat of the melting was 26.02 J/g.

The percent crystallinity was determined using the following equation [39]:(10)$$\%Crystallinity=\frac{\Delta H_{m}-\Delta H_{c}}{\Delta H_{m}{}^{\circ}}\cdot 100\%$$
where $\Delta H_{c}$ is the latent heat of crystallization and $\Delta H_{m}$ is the latent heat of melting; $\Delta H_{m}{}^{\circ}$ is a reference heat of melting for a 100% crystalline polymer (for a 100% crystalline PLA $\Delta H_{m}=93\;J/g)$ [39,40].

Based on the ratio from [40] of the energy effects of the exo- and endo-peaks, it can be assumed that %Crystalinity=2.2% and the polymer is predominantly amorphous, which confirmed the information from the supplier.

### 3.2. Uniaxial Tensile Test

A total of 480 samples were tested for tension. For each sample, a strain–stress curve was obtained, which is the maximum value of the stress that the sample experiences until the moment of fracture. The test results were processed by the methods of statistical analysis (see Section 3.3) using Statistica (ver. 13). Table 2 consists of the tensile strength values for the top ten sample printing modes that were of greatest interest.

Conditionally, they can be divided into three groups: modes with the best mechanical properties, modes with the worst mechanical properties, and the two modes that were recorded as outliers on graphs when performing the ANOVA analysis. Moreover, the nature of the behavior of the properties in these two modes did not correspond to the general logic of the behavior of the other samples. These specimens were selected to be tested with the shape memory cycle.

According to Table 2, the highest values of the tensile stress were shown by samples with a print temperature of 240 °C and a nozzle diameter of 0.5 mm while the lowest values were for samples with a print temperature of 210 °C and a nozzle diameter of 0.3 mm. A sample with 80% filling and line pattern provided the highest ultimate tensile strength. This sample had a maximum tensile strength of 50.1 MPa. As can be seen from the table, the same line drawing with 50% filling, but already with a nozzle temperature of 210 °C, nozzle diameter of 0.3, and a layer height of 0.3 mm, showed the lowest ultimate tensile strength of 32.9 MPa. The study in [41] showed that the value of the tensile strength for a linear pattern was greater than for grid filling, which, by and large, was implicit in our study regarding the presence of more significant parameters. In a study by Corapi et al. [42], the tensile strength of the PLA sample ranged from 28 to 58 MPa, depending on the orientation of the print. Rao et al. [43] estimated the effect of the layer height in the PLA sample on the tensile strength, which was in the range from 21 to 26 MPa. The strength values obtained by us during the experiment corresponded to the reality and were in the upper limits for this material.

Below, we demonstrate the outcomes of the Ansys numerical approach on a single specimen out of 480. Figure 3 shows the simulation of the dog-bone specimen for a tensile test based on the experimental data for printing regimen no. 17, which presented the highest tensile strength. The difference between the experimental and simulation data is presented in Figure 4.

Thus, the results of the tensile test simulation were in good agreement with the experimental data and confirmed the adequacy of the selected Mooney–Rivlin model.

### 3.3. Analysis of Variance

The ANOVA analysis was performed in this study to determine the degree of influence of one or more parameters on the sample’s tensile strength. This allowed us to determine the most notable printing regimes for the further thermomechanical study after examining the dependencies. Statistica software (ver. 13) was used to conduct the analysis. The tensile strength was chosen as the output parameter, while all five printing parameters were chosen as inputs. As a result, the multivariate analysis of variance was used to determine the impact of numerous factors on the dependent variable.

First, each parameter’s independent impact on the stress value was analyzed. The dependence of strength on parameters such as nozzle diameter and extrusion temperature, for which the *p* value was less than 0.05, is shown in Table 3 and Table 4. The same data in terms of plots are presented in Appendix A.

The difference between the average values of the factor is significant if the *p* value is small (*p* < 0.05). As a result, these two factors had a large influence on the dependent variable. The primary assumption was that a greater value of the nozzle diameter corresponded to a greater stress value and the higher the temperature of the extruder, the greater the strength of the part we observed. Whether everything was so unambiguous was found out during our evaluations. Considering other printing parameters such as the layer height, infill pattern, and density, in our research, their single impacts on the tensile strength were too insignificant with a *p*-value >> 0.5.

The next step was to consider the combined influence of factors. First, we studied the combined effects of two factors. Among the 10 combinations, only one showed statistically significance—extrusion temperature and layer height. Next, we continued to increase the number of factors for the total influence assessment. The mutual influence of two and three parameters on the tensile strength is presented in Table 5 and Table 6. The same data in in terms of plots are presented in Appendix A.

Here, we focused on the parameters that had previously been highlighted as being significant. However, since there are so many parameters and determining how they interact is a complex challenge, we summarized the results of our analysis for a further selection and study of the samples. It can be assumed that the nozzle diameter of 0.5 mm corresponded to higher values of maximum stress, and the temperature of the extruder significantly affected the result. The smaller the ratio of the layer height and diameter, the better for the strength properties in the case when the temperature was 210 °C. The reverse situation was typical for a temperature of 240 °C.

### 3.4. Dynamic Mechanical Analysis

The printing modes shown in Table 7 were selected and studied for DMA based on the ANOVA results. The choice of the tested samples can be explained by the versatile combinations of the printing parameters and different values of mechanical strengths. We selected two samples, no. 10 and no. 17, with the highest strength values, and two samples no. 113 and no. 117, with the worst. Sample no. 2 showed average stress values under uniaxial tension.

We tried to consider critical cases to expect different data, since there was the assumption that the DMA results for one material would be similar despite the print parameters. One essential result of DMA was the glass transition temperature, which corresponded to a rapid fall in the storage modulus or a peak in the tan delta and loss modulus curve. The behavior of the storage modulus, loss modulus, and tan delta as a function of temperature, obtained as a result of the measurements, is depicted in Figure 5.

The glass transition temperatures obtained by these three curves varied by up to 5 °C. In addition, we observed a 10 °C deviation from the DSC defined glass transition temperature (Figure 2). It is crucial to describe how the T_g_ should be obtained and in which experimental conditions. The experimental results are shown in Table 8.

The glass transition temperature was in the range of 63–64 °C from the storage or loss modulus and 69 °C determined from the tan delta. The T_g_ value for each sample varied within 1%, which means that this is an acceptable measurement error, and its value is unique for samples from the same material.

Experimental data were used to fit the curve specified in Equation (5), which allowed us to determine the components of the Prony series, which are necessary to use the viscoelastic model in Ansys. The number of Prony terms (relaxation time and modulus) can be any; the only condition is the convergence of the curves. In this study, we tried to use only two terms, but the convergence was far from ideal, after which we experimentally increased the number of terms to N=8. In our case, Ansys requests a table with the $c_{i}$ terms representing the normalized Prony coefficients for the shear behavior, and the $\beta_{i}$ values represent the relaxation times of the Prony series. Figure 6 contains graphs showing the result of fitting the curve to the experimental points.

In a section up to 50 °C, the fitted curve had a non-uniform representation in the form of steps. Increasing the Prony series to eight arguments led to a decrease in the move of these steps, but it was not possible to achieve a fully linear representation of this section. Increasing the number of arguments up to nine, convergence was not observed. The obtained results were evaluated as optimal. Each curve corresponded to its own set of Prony series.

The experimental results of DMA were used to extract the viscoelastic characteristics, which were subsequently used as defining parameters of the model in the simulation. The modulus of elasticity was calculated from the curve of the storage modulus according to Equation (5). In addition, the values of the coefficient of thermal expansion at room temperature and at the glass transition temperature were obtained during thermomechanical analysis. Here, we used the CTE values for the normal direction and the second heating curve from Table 1. Table 9 contains the data used in Ansys simulation to define the PLA material model.

As output values from the Ansys simulation, the amplitude of strain and stress were successfully obtained. Every temperature point starting from 30 °C to 80 °C with an increment of 5–10 °C (near T_g_ 1 °C) was simulated and had its own value of strain and stress. The phase shift δ was taken from the experimental tan(δ) data. For each temperature point, these values were substituted into the corresponding formula (Equation (9)) to calculate the storage module (Equation (5)). All calculations were performed and save in an Excel file and then exported to Origin 2021 software. To compare the simulation and the experiment, we plotted two curves of each sample on the same graph. The temperature dependence of the real part of the complex module was determined during the experiment and as a result of the numerical solution, is shown in Figure 7.

Consider the printing regimes no. 2 and no. 17, where the difference in parameters was the value of the layer height (0.2 mm and 0.3 mm, respectively). At the starting point, the storage modulus experimental value was 2188 MPa for sample no. 2 (2070 MPa for sample no. 17) and the modeling value was 2064 MPa for sample no. 2 (2337 MPa for sample no. 17). As the free volume continued to increase with the increasing temperature, the glass transition occurred. The onset of the storage modulus curve drop was used to detect this transition point. The experimental curve had a step at T_g_ = 64.22 °C for sample no. 2 (63.51 °C for sample no. 17) and the simulation had a step at T_g_ = 63 °C for sample no. 2 (62.8 °C for sample no. 17). The experimental storage modulus value at T_g_ was 859.6 MPa and the simulation value was 1029 MPa for sample no. 2. The sample with printing mode no. 17 had an experimental storage modulus equal to 1211 MPa, and the simulation curve showed 1314 MPa. Thus, we received the overestimated storage module values per 103–170 MPa. The curve also showed the softening process at the glass transition, where the modulus fell to less than 10 MPa. The storage modulus changed by about two orders during the glass transition. The area after the T_g_ had a rubbery plateau. The modulus in the plateau region was 1 MPa for sample no. 2 (6 MPa for sample no. 17).

Printing regimes no. 113 and no. 117 had common parameter values (nozzle diameter 0.3 mm, layer height 0.3 mm, extruder temperature 210 °C) and differed in filling (no. 113—lines 50%, no. 117—gyroid 25%). These samples showed the worst mechanical tensile properties. In the very beginning (T = 30 °C), the experimental storage modulus was 2264 MPa for sample no. 113 (2056 MPa for sample no. 117) and the modeling value was 2414 MPa for sample no. 113 (2155 MPa for sample no. 117). The glass transition temperature was T_g_ = 63.69 °C for sample no. 113 (63.50 °C for sample no. 117) and from the modeling curve, the T_g_ = 63 °C for sample no. 113 (63 °C for sample no. 117). The storage modulus at the T_g_ point in the experiment was 1064 MPa and the simulation value was 1071 MPa for sample no. 113. For sample no. 117, the experimental storage modulus was equal to 1078 MPa and the simulation curve was 1085 MPa. The modulus in the plateau region was 5 MPa for both. In the cases discussed, we connected the difference between the experimental and numerical values of the storage modulus with the need to further improve our numerical model.

The apparent difference between the numerical and experimental curves can be explained by the insufficient correctness of the temperature–time analogy used in the calculation when passing through the glass transition interval. It is important to note that the values of the physical characteristics that were obtained experimentally are not truly accurate; various factors of influence must be taken into account, for example, the interaction of the sample with the grips of the testing machine. The causes and elimination of inaccuracies require further research.

### 3.5. Thermomechanical Shape Memory Cycle

SMPs showed excellent performance at temperatures below and above the glass transition temperature T_g_. The shape recovery ratio indicates the ability of the SMP to recover its original shape from a temporary or deformed shape. The shape recovery and shape fixity ratios for different printing regimes and five cycles per each are presented in Table 10 and Table 11, respectively.

The shape memory cycle was repeated up to five times for each sample in order to assess the effect of fatigue on the characteristics of the SMP. The deformation remained constant at 60% relative to the original gauge length. Figure 8 illustrates the specimen with printing mode no. 17 before and after the shape memory cycles.

The fixation of the shape practically did not change during the entire experiment and was close to 100%. Another situation arose when assessing the ability to recover the shape, which, even after the first cycle, was not comparable to the maximum 100%, and each cycle deteriorated significantly. Printing modes no. 10, no. 15, no. 16, and no. 17 showed the best mechanical tensile properties (tensile strength 49–50 MPa). All samples were printed at an extruder temperature of 240 °C and a nozzle diameter of 0.5 mm. When the density changed from 20% to 25% (printing modes no. 16 and no. 17, respectively), the coefficient of shape recovery for the first cycle increased from 84.4% to 93.1%, and for the last cycle decreased from 45.5% to 61.3%, accordingly. The average value of form fixation for all cycles was 98.76% for no. 16 and 98.62% for no. 17. With different line and gyroid filling patterns (printing modes no. 10 and no. 15, respectively) and the same other parameters, the coefficient of shape recovery for the first cycle was 91.4% for the line pattern (no. 10) and only 75% for the gyroid (no. 15). After five cycles, the recovery ratios changed to 71.6% and 42.1%, respectively. The average fixation coefficient for all cycles of sample no. 10 turned out to be 99.38%, and for sample no. 15, it was 98.88%. SME in the PLA after five cycles of stress–strain training in comparison with the first cycle is shown in Figure 1. We can conclude that the density and method of filling the sample volumes during printing significantly affected the appearance of the SME in the polymer. However, these manifestations were multidirectional.

The shape fixity and recovery for the specimens with the highest tensile strength are shown in Figure 9.

Information about the regimes that showed the worst mechanical properties (tensile strength 32–35 MPa) is presented in Figure 10.

The following printing values were common for all modes (no. 107, no. 113, no. 115, no. 117): nozzle diameter 0.3 mm, layer height 0.3 mm, extruder temperature 210 °C. Comparing the two modes, no. 107 and no. 117, one with a grid filling of 25%, and the other with a gyroid of 25%, respectively, we can say that the coefficient of shape recovery for the first cycle was 73% and 79.2% and after five cycles, their values changed by 51.2% and 61.7%, respectively. For both modes, the average fixation coefficient was 98%. Samples no. 113 and no. 115 differed only in density (lines 50% for no. 113, lines 80% for no. 115). The coefficient of form recovery after the first cycle was 78.2% and 76%, and after the fifth cycle, it was 55.3% and 29%, respectively. For both modes, the average fixation coefficient was around 99%.

The shape memory and shape fixity coefficients of printing modes no. 2 and no. 45 are presented in Figure 11.

Samples with printing modes no. 2 and no. 45 (tensile strength 43–44 MPa) were not in the categories with the best or worst mechanical characteristics. They aroused interest at the ANOVA analysis stage because they had an atypical behavior among other modes (see Section 3.3). Sample no. 2 (nozzle diameter 0.5 mm, layer height 0.2 mm, with a mesh filling of 25% and an extruder temperature of 240 °C) showed recovery after the first cycle by 86.1%, and after the fifth cycle by 59.9%. The average fixation of the form was 97.4%. Sample no. 45 (nozzle diameter 0.5 mm, layer height 0.2 mm, with gyroid filling 80%, and extruder temperature 210 °C) showed recovery after the first cycle by 74.6%, and after the fifth, it was 41.8%. The average fixation of the form was 98.9%.

The results of the stress–strain–temperature curve in 3D space for four different samples after the first cycle are shown in Figure 12. As expected, the stress–strain curve followed a linear trend for the first step (1), with a maximum strain value of 60%. As the temperature decreased, residual stresses accumulated due to the interaction of thermal and mechanical energy (2). The third stage showed stress relaxation and the ability of the material to memorize the shape, since the restriction is removed at low temperature (3). The shape was restored only after the temperature exceeded 64 °C (4). The deformation after the shape recovery stage was fixed at a level of more than 20%. The maximum stress value for mode no. 2 was 1.68 MPa, and the value of the final deformation was 0.23 mm/mm.

Printing regimes no. 2 and no. 17 only showed a difference in the layer height (0.2 mm and 0.3 mm, respectively). Other parameters such as nozzle diameter 0.5 mm, infill line 80%, and an extruder temperature of 240 °C were the same for both. The coefficients of shape recovery were 86.1% and 93.1% due to the differing residual stresses of 1.68 MPa and 1.72 MPa, respectively. Two printing regimes, no. 113 and no. 117, with a difference in the infill and density (line 50% and gyroid 25%) had the same nozzle diameter of 0.3 mm, layer height of 0.3 mm, and an extruder temperature of 210 °C. The values of their shape recovery ratios were equal to 78.2% and 79.2%, respectively. The residual stresses rose to 1.87 MPa and 1.06 MPa, and the values of the final strain were 0.218 and 0.208 mm/mm, respectively.

## 4. Discussion

The results reported here confirm that the initial parameters of the printing mode directly influence the mechanical performance. From the ANOVA, we observed that the contribution of the extrusion temperature and nozzle diameter was the maximum compared to the other process parameters. However, it is important to emphasize that when talking about the diameter of the nozzle, it is necessary to consider the height of the layer. The experimental maximum in tensile strength of 50.1 MPa was observed for the 0.5 mm nozzle diameter, 0.2 mm layer thickness, line infill with 80% density raster, and temperature of 240 °C. The lowest tensile strength values were found in samples with printing modes in which the layer height and nozzle diameter were equal, and the extruder temperature was 210 °C.

We suppose that the number and size of the voids formed between the strands had a significant impact on the total strand adhesion. While the expected finding was presented in [44], this study showed that when the ratio between the layer height and the nozzle diameter decreased from 1 to 0.125, the porosity decreased from about 20% to 3%. In our study, this ratio took values from 1 to 0.4, where 1 corresponds to modes with a nozzle diameter D = 0.3 mm and a layer height H = 0.3 mm, and a ratio of 0.4 corresponds to modes with D = 0.5 mm and H = 0.2 mm. Therefore, for modes with a layer height of 0.2 mm and 0.3 mm with a nozzle diameter of 0.3 mm, the porosity was much higher than for modes with a nozzle diameter of 0.5 mm. This means that an influence of the ratio of these two parameters on mechanical performance was observed.

The experimental strain–stress curve and the identical simulation curve exhibited a similar trend with low inaccuracy when they were compared. On this basis, it can be assumed that the Mooney–Rivlin model with five parameters used to represent the PLA material in the simulation corresponded well with the actual data, implying that the model can predict and recreate the material’s real mechanical response to the applied uniaxial tensile load. The same conclusion was reached by Meng et al. in his study of the tensile properties of polymers [45].

The obtained DSC results show the energy effects of the exo- and endo-peaks, the ratio of which it can be concluded that the samples under study were predominantly amorphous. This means that the transition point between the phases was characterized by the glass transition temperature, the value of which had also been estimated as T_g_ = 55 °C. Based on the obtained TMA information, it was interesting to see how the thermal expansion of the sample changed in the temperature range before the transition to the rubber state. In the glassy state, the CTE was about 70–80 ppm/K only, and above the glass transition temperature, it was 250–270 ppm/K. Moreover, the significantly anisotropic nature of the CTE behavior was discovered and its changes at the stages of heating and cooling were not obvious.

The DMA experiment showed that, regardless of the configuration of the printing process, the values of the glass transition temperature of the elasticity modulus at the initial stage of the curve were in the same range, and did not differ from each other by a value greater than 1–2%. The glass transition temperature for all samples with different measurement curves ranged from 63 to 69 °C. This is quite acceptable, given that the values of the glass transition temperature obtained within the same curve (for example, the curve of the storage module) differed within 1 °C. The output values confirmed our assumption that the DMA was not sensitive to the internal parameters of the FDM part, and that only the material under study is important. Similar conclusions were drawn by Bopp and Behrendt in [46], who demonstrated that frequency dependence in DMA is constant across a wide range of parameter settings.

When comparing the simulation curve with the experimental curve, a noticeable difference in the values of the elastic modulus could be observed. The reason for this could be an inaccurate reconstruction of the PLA material model in the simulation environment. Thus, the model requires further clarification. We assume that in order to obtain a more accurate curve, we could use a Prony series from a possible relaxation experiment, which was not conducted in our research.

Thermocycle tests showed that the samples had a SME. For this study, the samples were divided into groups with the best mechanical and worst mechanical properties as well as those modes that showed average strength values, but did not fall into the typical trend of the ANOVA curve. Sample no. 10 (nozzle diameter 0.5 mm, layer height 0.2 mm, with a line filling of 80%, and an extruder temperature of 240 °C) showed one of the highest tensile strengths (50.1 MPa) and the best values of the shape recovery coefficient for five cycles. There was a slight deterioration from cycle to cycle, but its drop was significantly less than in other samples (2–6%). The worst values were demonstrated by sample no. 115 (nozzle diameter 0.3 mm, layer height 0.3 mm, with a line filling of 80%, and an extruder temperature of 210 °C). Since these two samples (no. 10 and no. 115) showed high and low values in tensile strength, respectively, we can believe that the strength featured had an effect on the shape recoverability. Speaking of the fixity ratio, with each cycle, it decreased by 0.1–1%; for all samples, the values were close to 100%.

## 5. Conclusions

This study presented mechanical and thermomechanical tests aimed to predict the SMP behavior in PLA printed samples, especially in the range of glass transition temperature. In addition to evaluating the mechanical tensile strength, DSC, TMA, DMA, and the shape memory tests for five cycles were investigated.

We obtained results showing that the use of different printing parameters in sample production affects the tensile strength. It has been shown that samples with an extrusion temperature of 240 °C and a nozzle diameter of 0.5 mm had the highest values of tensile strength, while the lowest values were observed with an extrusion temperature of 210 °C and a nozzle diameter of 0.3 mm. The Mooney–Rivlin hyperelastic model with five parameters resulted in a good match between the simulation and the experimental data.

For the first time for the PLA material and FDM approach, the TMA experiment allowed us to evaluate the features of the thermal deformation of the samples to obtain the CTE values at different temperatures, space directions, and running curves. Experimentally obtained CTEs showed strong anisotropy and were successfully used in the material description during DMA simulation in Ansys. Based on the results of the DSC experiment, we were convinced of the amorphous structure of the sample, and determined the glass transition temperature as the phase transition point for the SME.

The FEM was implemented to simulate the dynamic response of the samples in a temperature regime. The viscoelastic characteristics of the PLA material were obtained by fitting the experimental curve to the Prony series, the time–temperature superposition method was applied. A comparison between the experiment and simulation curves showed that the values of the storage modulus in the range of glass transition temperature varied, but the curves showed a similar nature.

Our study highlighted that PLA really has a SME. Moreover, we noticed the influence of the mechanical characteristics of the samples on its ability to recover the shape. Thus, the stronger the sample, less fatigue from cycle to cycle was observed when restoring the initial shape after deformation. A significant decrease in the recovery ratio from cycle to cycle can be explained by the too large value of the applied deformation, despite the shape fixity, which did not decrease with each SMP cycle.

Future efforts should aim to create a polymer model with the SME, which will include elastic, viscoplastic, and thermal deformation, and is likely to contribute to the correct prediction of the SMP behavior. The final goal of such endeavors is to combine all of the experimental and simulation data into a neural network as well as to build a digital twin of the FDM process for SMP materials.

## Figures and Tables

**Figure 1 polymers-15-01162-f001:**
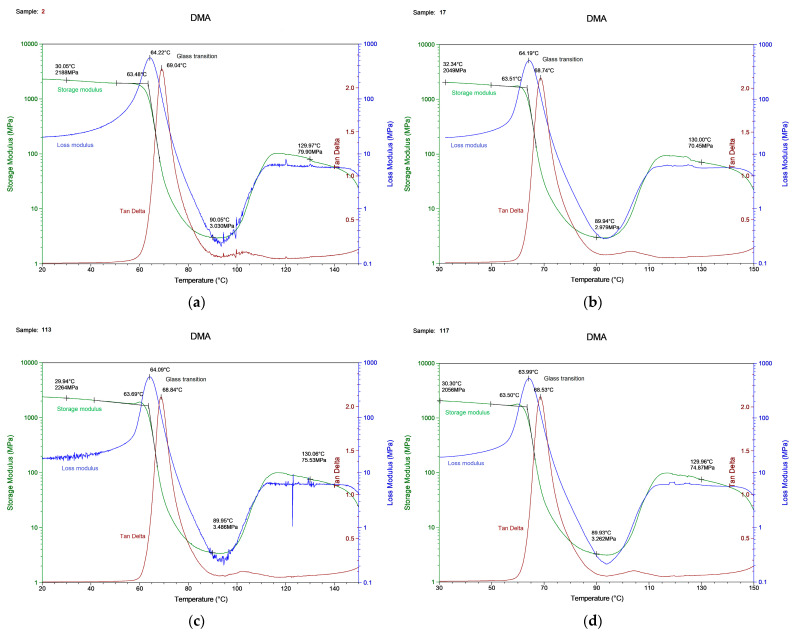
Specimen no. 17 at the maximum ε_yy_ displacement levels in the first and fifth cycles.

**Figure 2 polymers-15-01162-f002:**
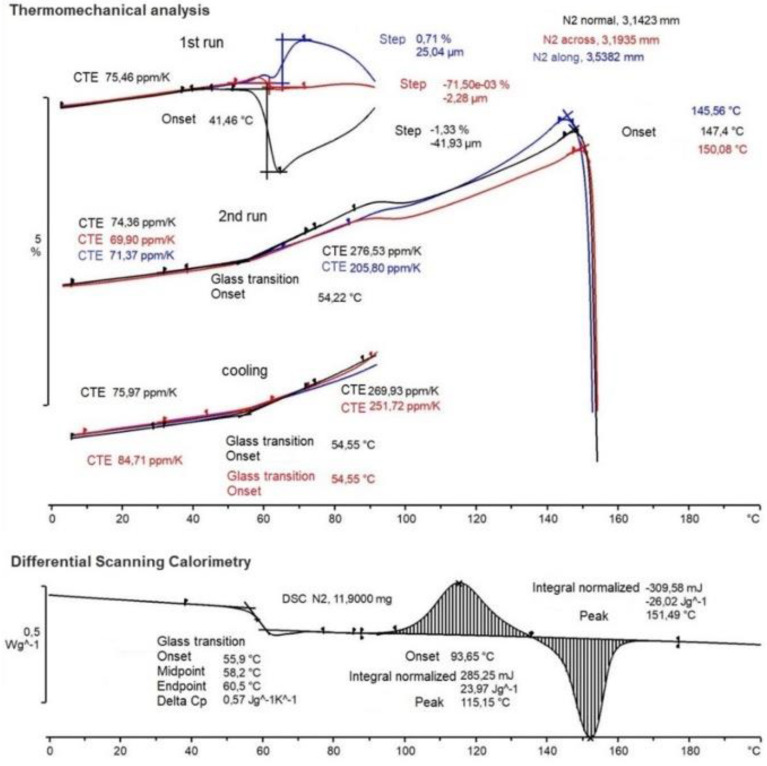
The TMA (top) and DSC (bottom) experimental results.

**Figure 3 polymers-15-01162-f003:**
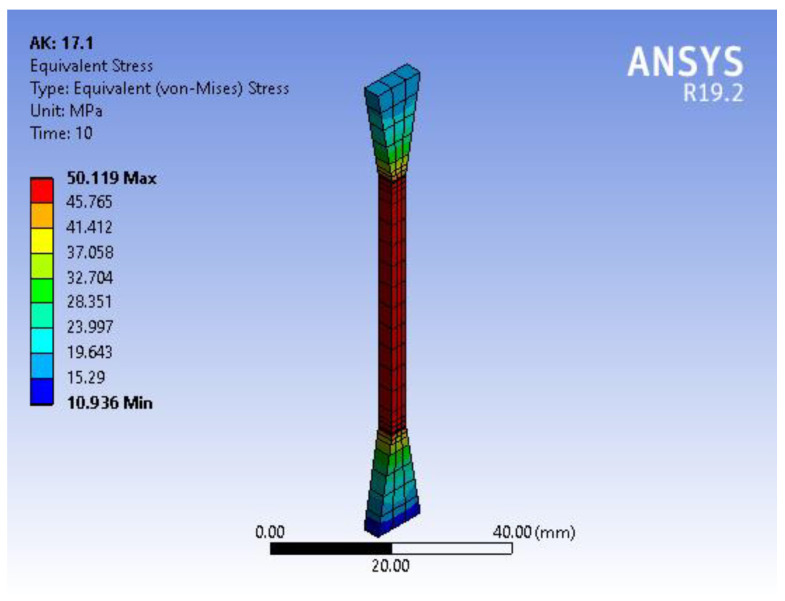
Tensile test simulation.

**Figure 4 polymers-15-01162-f004:**
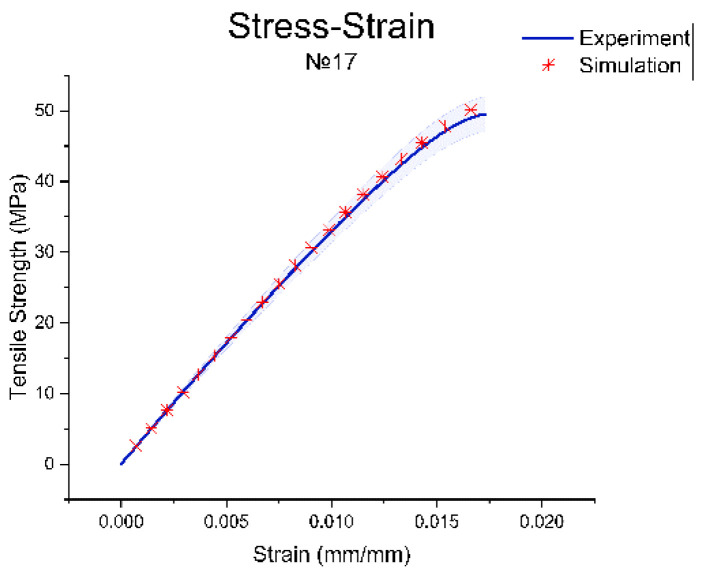
Experimental and simulation curves.

**Figure 5 polymers-15-01162-f005:**
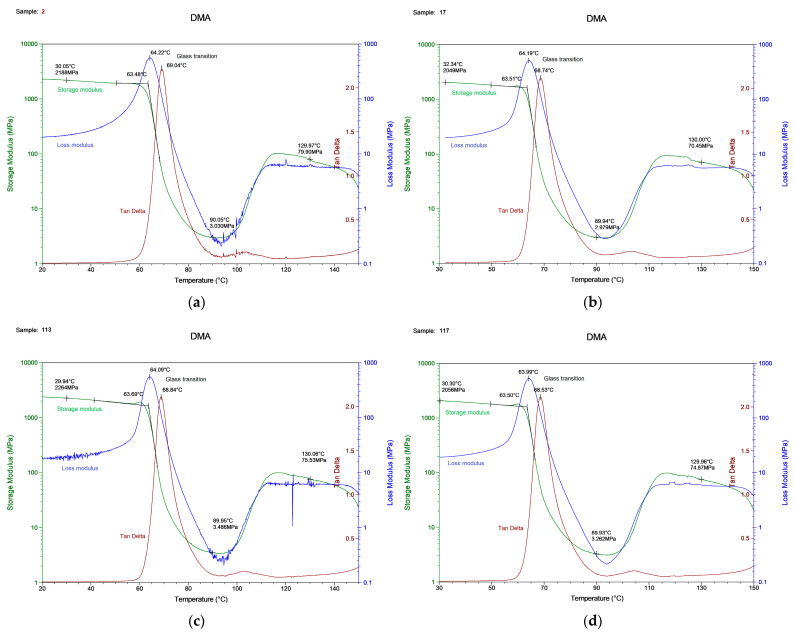
The DMA results for the (**a**) specimen with printing regime no. 2, (**b**) no. 17, (**c**) no. 113, (**d**) no. 117.

**Figure 6 polymers-15-01162-f006:**
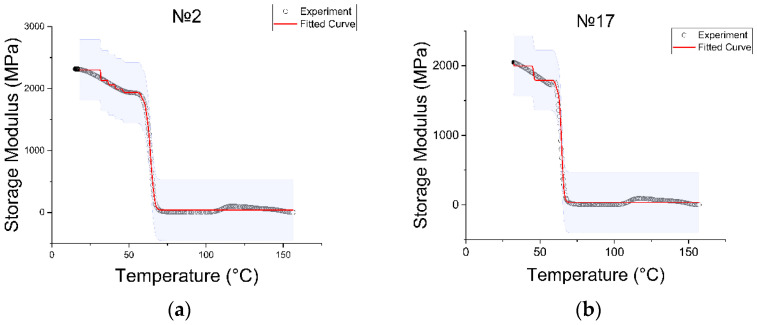
Convergence of the experimental and fitted curves of the specimen: (**a**) specimen with printing regime no. 2, (**b**) no. 17, (**c**) no. 113, (**d**) no. 117.

**Figure 7 polymers-15-01162-f007:**
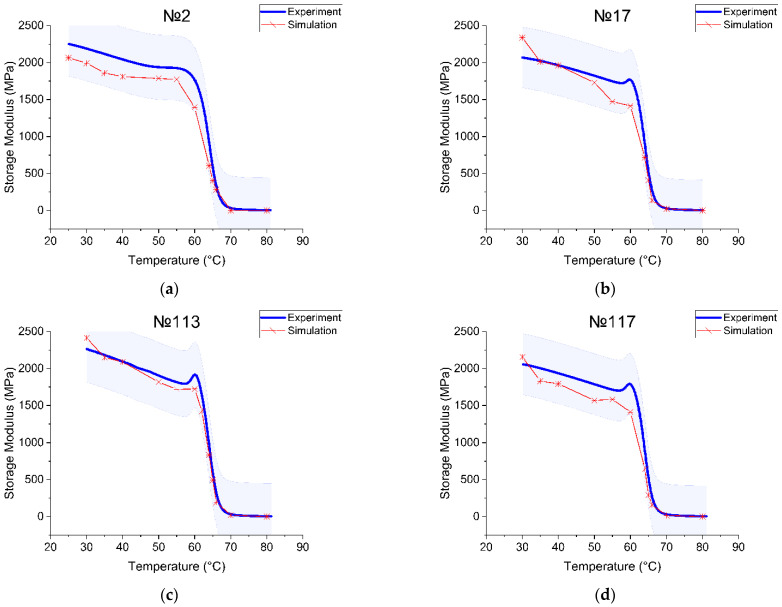
Comparison between the experimental results and simulation for the (**a**) specimen with printing regime no. 2, (**b**) no. 17, (**c**) no. 113, (**d**) no. 117.

**Figure 8 polymers-15-01162-f008:**
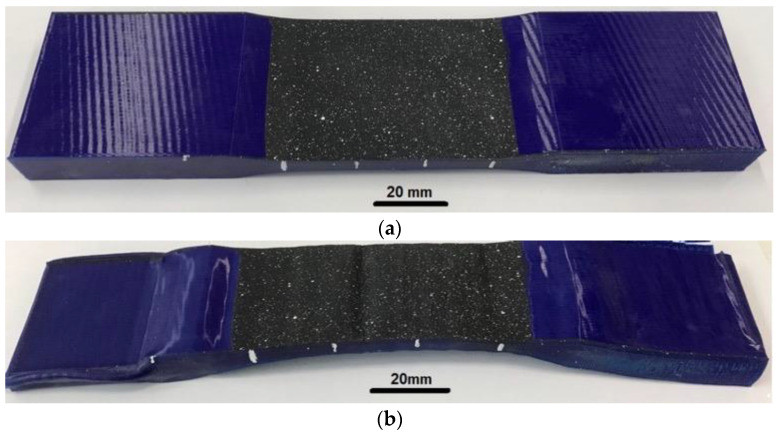
The specimen no. 17: (**a**) before the shape memory test; (**b**) after 5 SMP cycles.

**Figure 9 polymers-15-01162-f009:**
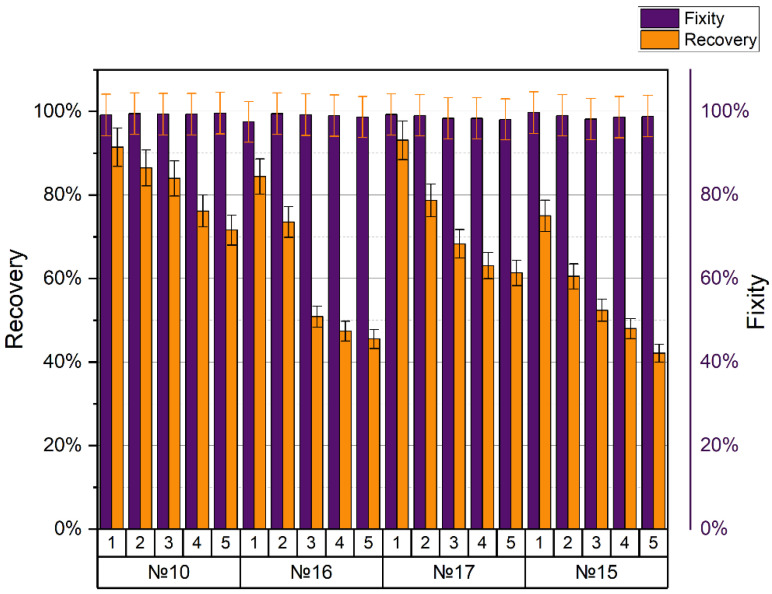
Shape recovery and fixity for the specimens with the highest tensile strength.

**Figure 10 polymers-15-01162-f010:**
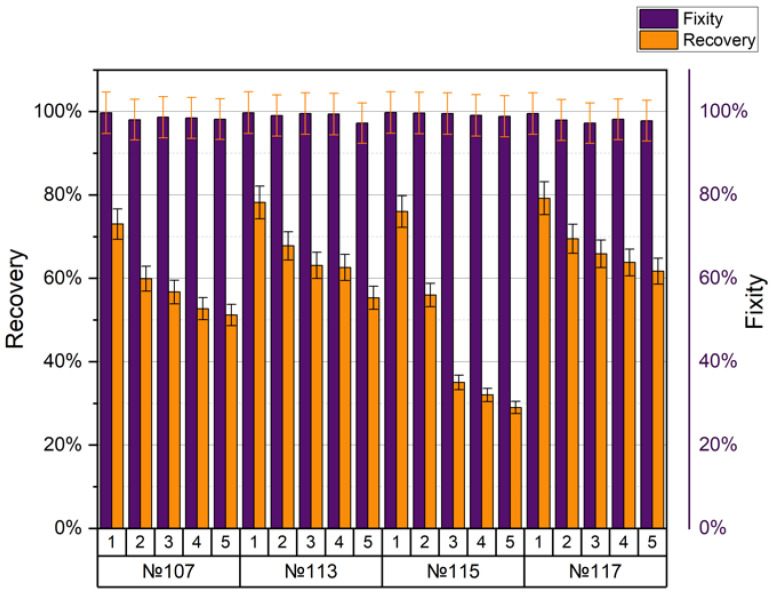
Shape recovery and fixity for the specimens with the lowest tensile strength.

**Figure 11 polymers-15-01162-f011:**
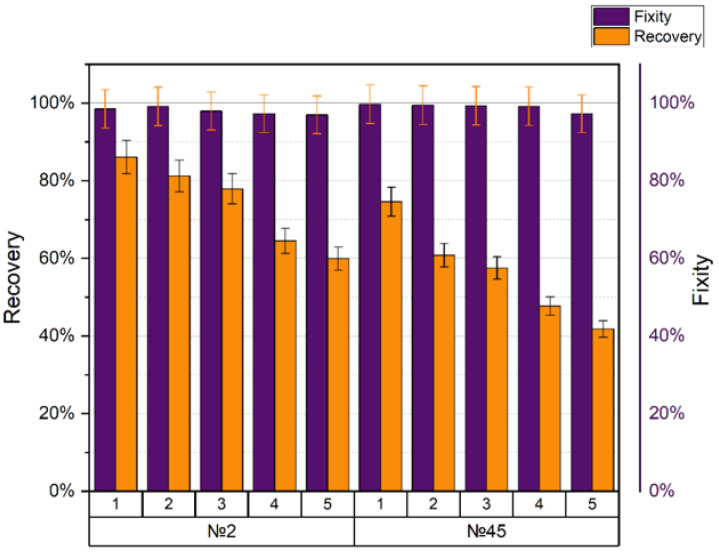
Shape recovery and fixity for specimens with ambiguous behavior.

**Figure 12 polymers-15-01162-f012:**
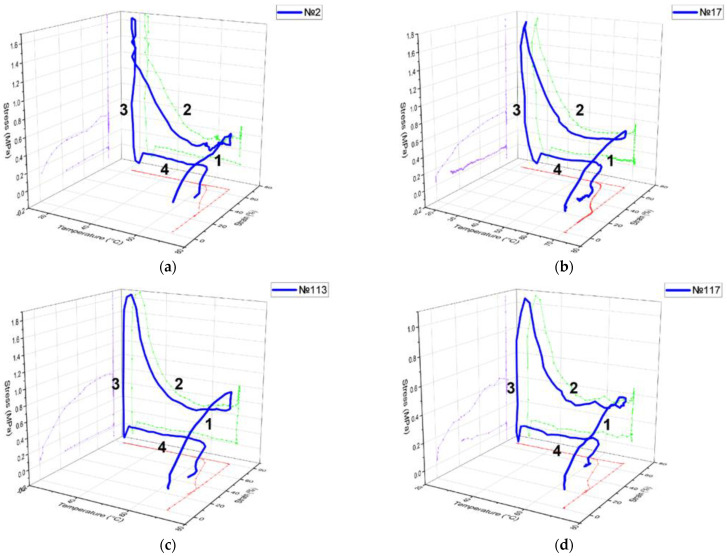
Shape memory baseline for the (**a**) specimen with printing regime no. 2, (**b**) no. 17, (**c**) no. 113, (**d**) no. 117.

**Table 1 polymers-15-01162-t001:** Coefficients of thermal expansion for different cases.

Direction	Glassy State	Above T_g_
First heating curve		
Normal (Z)	75.46 ppm/K	-
Across (X)		
Along (Y)		
Cooling curve		
Normal (Z)	75.97 ppm/K	269.93 ppm/K
Across (X) Along (Y)	84.71 ppm/K	251.72 ppm/K
Second heating curve		
Normal (Z)	74.36 ppm/K	276.53 ppm/K
Across (X)	69.9 ppm/K	205.8 ppm/K
Along (Y)	71.37 ppm/K	

**Table 2 polymers-15-01162-t002:** Specimens selected for the shape memory cycle test.

Specimen	D of the Nozzle	Layer Height	Infill Pattern	Infill Density	Extrusion Temperature	Tensile Strength
Specimens with the highest tensile strength values						
10	0.5 mm	0.2 mm	Line	80%	240 °C	50.1 MPa
16	0.5 mm	0.3 mm	Grid	20%	240 °C	49.9 MPa
17	0.5 mm	0.3 mm	Grid	25%	240 °C	49.8 MPa
15	0.5 mm	0.2 mm	Gyroid	80%	240 °C	49.1 MPa
Specimens with the lowest tensile strength values						
113	0.3 mm	0.3 mm	Line	50%	210 °C	32.9 MPa
117	0.3 mm	0.3 mm	Gyroid	25%	210 °C	34.2 MPa
115	0.3 mm	0.3 mm	Line	80%	210 °C	34.9 MPa
107	0.3 mm	0.3 mm	Grid	25%	210 °C	35.2 MPa
Samples with ambiguous behavior						
2	0.5 mm	0.2 mm	Grid	25%	240 °C	43.2 MPa
45	0.5 mm	0.2 mm	Gyroid	80%	210 °C	44.5 MPa

**Table 3 polymers-15-01162-t003:** Descriptive statistics for the nozzle diameter factor.

Nozzle Diameter	Mean (MPa)	Standard Deviation
0.5 mm	46.87	0.45
0.3 mm	38.31	0.45

**Table 4 polymers-15-01162-t004:** Descriptive statistics for the extrusion temperature factor.

Extrusion Temperature	Mean (MPa)	Standard Deviation
240 °C	43.18	0.45
210 °C	41.99	0.45

**Table 5 polymers-15-01162-t005:** Descriptive statistics for the combined influence of the extrusion temperature and layer height.

Extrusion Temperature	Layer Height	Mean (MPa)	Standard Deviation
240 °C	0.3 mm	43.69	0.64
240 °C	0.2 mm	42.67	0.64
210 °C	0.2 mm	42.51	0.64
210 °C	0.3 mm	41.47	0.64

**Table 6 polymers-15-01162-t006:** Descriptive statistics for the combined influence of the extrusion temperature, nozzle diameter, and layer height.

Nozzle Diameter	Extrusion Temperature	Layer Height	Mean (MPa)	Standard Deviation
0.5 mm	240 °C	0.3 mm	47.47	0.9
0.5 mm	240 °C	0.2 mm	47.06	0.9
0.5 mm	210 °C	0.2 mm	46.53	0.9
0.5 mm	210 °C	0.3 mm	46.39	0.9
0.3 mm	240 °C	0.3 mm	39.92	0.9
0.3 mm	210 °C	0.2 mm	38.48	0.9
0.3 mm	240 °C	0.2 mm	38.27	0.9
0.3 mm	210 °C	0.3 mm	36.55	0.9

**Table 7 polymers-15-01162-t007:** Specimens and printing modes selected for DMA.

No.	Printing Regime	D of the Nozzle	Layer Height	Infill Pattern	Infill Density	Extrusion Temperature	Tensile Strength
1	10	0.5 mm	0.2 mm	Line	80%	240 °C	50.1 MPa
2	17	0.5 mm	0.3 mm	Grid	25%	240 °C	49.8 MPa
3	2	0.5 mm	0.2 mm	Grid	25%	240 °C	43.2 MPa
4	117	0.3 mm	0.3 mm	Gyroid	25%	210 °C	34.2 MPa
5	113	0.3 mm	0.3 mm	Line	50%	210 °C	32.9 MPa

**Table 8 polymers-15-01162-t008:** The DMA results.

Specimen	T_g_ (Storage Modulus)	T_g_ (Loss Modulus)	T_g_ (Tan Delta)	Storage Modulus at T = 30 °C	Storage Modulus at T_g_	Loss Modulus at T = 30 °C	Loss Modulus at T_g_
2	64.22 °C	64.09 °C	69.04 °C	2188 MPa	859.6 MPa	20.37 MPa	555 MPa
17	63.51 °C	64.22 °C	68.74 °C	2049 MPa	1051 MPa	20.04 MPa	543.8 MPa
113	63.69 °C	64.19 °C	68.84 °C	2264 MPa	1064 MPa	19.02 MPa	553.8 MPa
117	63.50 °C	63.99 °C	68.53 °C	2056 MPa	1078 MPa	19.36 MPa	556.7 MPa

**Table 9 polymers-15-01162-t009:** Data of the temperature dependent material from the TMA and DMA data.

Temperature	Tensile Moduli	Poisson’s Ratio	Thermal Expansion Coefficient [Table 1]
22 °C	2300 MPa	0.36	74.36 ppm/K
64 °C	929 MPa	0.42	276.53 ppm/K

**Table 10 polymers-15-01162-t010:** The experimental shape recovery ratio for different specimens.

Specimen	R_rec_ 1 Cycle	R_rec_ 2 Cycles	R_rec_ 3 Cycles	R_rec_ 4 Cycles	R_rec_ 5 Cycles
2	86.1%	81.2%	77.9%	64.5%	59.9%
10	91.4%	86.5%	84.0%	76.2%	71.6%
15	75.0%	60.5%	52.4%	48.0%	42.1%
16	84.4%	73.5%	50.9%	47.4%	45.5%
17	93.1%	78.7%	68.3%	63.1%	61.3%
45	74.6%	60.8%	57.5%	47.7%	41.8%
107	73.0%	59.9%	56.7%	52.7%	51.2%
113	78.2%	67.8%	63.1%	62.6%	55.3%
115	76.0%	56.0%	35.0%	32.0%	29.0%
117	79.2%	91.0%	96.4%	97.1%	96.5%

**Table 11 polymers-15-01162-t011:** The experimental shape fixity ratio for different specimens.

Specimen	R_fix_ 1 Cycle	R_fix_ 2 Cycles	R_fix_ 3 Cycles	R_fix_ 4 Cycles	R_fix_ 5 Cycles
2	98.46%	99.14%	97.94%	97.22%	96.96%
10	99.17%	99.47%	99.33%	99.36%	99.58%
15	99.73%	99.05%	98.14%	98.63%	98.86%
16	97.48%	99.48%	99.21%	98.98%	98.66%
17	99.26%	99.05%	98.34%	98.33%	98.11%
45	99.74%	99.46%	99.31%	99.18%	97.21%
107	99.72%	98.06%	98.64%	98.46%	98.19%
113	99.74%	99.06%	99.55%	99.41%	97.22%
115	99.77%	99.66%	99.56%	99.10%	98.86%
117	99.56%	97.96%	97.22%	98.14%	97.80%

## Data Availability

The data presented in this study are available on request from the corresponding author after obtaining the permission of an authorized person.

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
