# Peer review of "Prediction of The Mechanical Behavior of Polylactic Acid Parts with Shape Memory Effect Fabricated by FDM"

_polymers, 2023, doi:10.3390/polym15051162_

Round 1

Reviewer 1 Report

In this paper, the mechanical behavior of PLA parts with shape memory effect fabricated by FDM process was investigated experimentally and numerically. The paper presents a high scientific content and important research aspects in the field of 4D printing.

1. The previous work is very low, and isn't sufficient. It is recommended updated this section with new references, and compared them with your paper.  There are many studies that address SMP.

2. What printer was used to make the samples?

3. Some manufacturing parameters should be added (Bed Temperature, Printing speed).  

4. “The deformation was applied to sections of small cross-section” a displacement of....mm was applied

5. Regarding FEA 

A. What simplifying hypotheses did you use in the FEA analysis (taking into account the anisotropy of the FDM process).

B. Mesh convergences should be involved, the optimal element and size element should be specified.

C.     Which element is used?

D.     What is the type of mesh?

6. The error that appears between the experimental data and the simulated data should be detailed. The authors should argue where this relative error comes from.

7. For the experimental data, a statistical analysis and tests to eliminate outliers should be performed.

Author Response

Reviewer 1

In this paper, the mechanical behavior of PLA parts with shape memory effect fabricated by FDM process was investigated experimentally and numerically. The paper presents a high scientific content and important research aspects in the field of 4D printing.

Response: Thank you for your questions, we appreciate your involvement.

  1. What printer was used to make the samples?

Response: The correction is made. Next information was added in the text. ‘Samples were fabricated using Picasso Designer X Pro 3D printer.’

  1. Some manufacturing parameters should be added (Bed Temperature 60, Printing speed). Response: The correction is made. Next information was added in the text. ‘The bed temperature is considered to be equal to 60 °C. The printing speed was set to 60 mm/s.’

  1. “The deformation was applied to sections of small cross-section” a displacement of....mm was applied.

Response: We followed your advice. ‘The displacement of 10*10-3 mm was applied to small cross-section of specimen.’

  1. Regarding FEA
  2. What simplifying hypotheses did you use in the FEA analysis (taking into account the anisotropy of the FDM process).

Response: We remember about the anisotropy of any AM process. We tried to choose the effective mesh size by iteratively refining the mesh until the results converged. Further mesh optimization resulted in time-consuming high-performance computing, which became unattainable in the end.

  1. Mesh convergences should be involved, the optimal element and size element should be specified.

Response: The preliminary convergence study with refined meshing allowed the appropriate mesh size to be set to 1 mm. The mesh was created using TET10 elements.

  1. Which element is used?

Response: The mesh was created using TET10 elements.

  1. What is the type of mesh?

Response: Quadratic tetrahedral mesh.

  1. The error that appears between the experimental data and the simulated data should be detailed. The authors should argue where this relative error comes from.

Response: The apparent difference between the numerical and experimental curves can be explained by the insufficient correctness of the temperature-time analogy used in the calculation when passing through the glass transition interval. It is important to note that the values of the physical characteristics that were obtained experimentally are not truly accurate; various factors of influence must be taken into account, for example, the interaction of the sample with the grips of the testing machine. We agree, that the causes and elimination of inaccuracies require further research.

  1. For the experimental data, a statistical analysis and tests to eliminate outliers should be performed.

Response: We are planning tests to eliminate outliers, and consider it a compulsory aspect in the continuation of our work. Statistical analysis ANOVA was performed in this study to determine influence of parameters on the sample's tensile strength.

We are sincerely grateful to the honorable Reviewer #1 for their careful and deep viewing of the paper, helpful comments and constructive remarks. We do believe that they really helped us to improve the paper and be more persuasive in reporting our results for the readers. We highly appreciate your time and efforts spent for editing the manuscript.

Reviewer 2 Report

The abstract is written very briefly. Most of it contains research methods. Almost half of the abstract is devoted to generalities and research methods. The abstract should be written more attractively. Also, the novelty of the article should be presented clearly. In addition, the conducted tests and their results should be added quantitatively and qualitatively.

Keywords can also be modified.

Investigation of mechanical properties, effect of printing parameters, and shape memory properties for PLA printed by FDM have been done abundantly in previous sources. What is the main difference between this research and previous sources?

Despite its large volume, the introduction is weak and superficial. Use new resources in the field of 4D printing by FDM to strengthen it. “A New Strategy for Achieving Shape Memory Effects in 4D Printed Two-Layer Composite Structures” --- “4D printing of PET-G via FDM including tailormade excess third shape” --- “4D PrintingEncapsulated PolycaprolactoneThermoplastic Polyurethane with High Shape Memory Performances”).

Figure 1 should either be deleted or completed by adding printed samples.

Add a scale bar to the figures.

Add the standards used in each part of the research method.

The standard deviation should be added to the results.

Table 1 should be deleted. Printing parameters are summarized in a table.

The existence of figures 2 and 3 is not required. Be omitted.

Section 2.5 should also be summarized. The equations in this section should be deleted. In this section, cooling rate, heating rate, load mode, frequency, temperature range, strain rate and ASTM should be presented.

The next section should be summarized.

Figures 4 and 5 should be merged.

Figures 6-8 should be deleted.

Figures 10-11 should be deleted.

Figures 12 and 13 for two variable printing parameters are meaningless. The results should only be presented in the table.

The conclusion is also long and should be corrected like the abstract.

How has the used model been verified? How are the structural equation and its parameters selected?

The research method section is long and should be summarized.

Author Response

Reviewer 2

The abstract is written very briefly. Most of it contains research methods. Almost half of the abstract is devoted to generalities and research methods. The abstract should be written more attractively. Also, the novelty of the article should be presented clearly. In addition, the conducted tests and their results should be added quantitatively and qualitatively.

Keywords can also be modified.

Response According Reviewer’s recommendations Abstract was rewrote. Thank you for your following comments, we appreciate your involvement.

  1. Investigation of mechanical properties, effect of printing parameters, and shape memory properties for PLA printed by FDM have been done abundantly in previous sources. What is the main difference between this research and previous sources?

Response Research on the mechanical properties and shape memory effect of PLA printed products is the focus of many scientific groups. In our study for the above-mentioned material, we not only obtained mechanical characteristics for different printing parameters, but also investigated the thermal behavior of FDM printed samples providing TMA+DSC as well as DMA analyses. Besides everything else, we investigated cyclic thermomechanical tests to study SME. To the best of the author's knowledge, in the first time for biocompatible PLA material and FDM approach, the results were brought together to create a more detailed portrait of PLA material and its behavior under different conditions. What is more, we were able to compare our experimental results of the tensile test and DMA with our FEM calculations.

  1. Despite its large volume, the introduction is weak and superficial. Use new resources in the field of 4D printing by FDM to strengthen it. “A New Strategy for Achieving Shape Memory Effects in 4D Printed Two-Layer Composite Structures” --- “4D printing of PET-G via FDM including tailormade excess third shape” --- “4D Printing‐Encapsulated Polycaprolactone–Thermoplastic Polyurethane with High Shape Memory Performances”).

Response: We followed your advice. According to recommendations of respected Reviewers #2 & #3 the Introduction section was rewrote.

  1. Figure 1 should either be deleted or completed by adding printed samples.

Response: Fig. 1 was deleted.

  1. Add a scale bar to the figures.

Response: Thanks a lot. Images of samples have a scale bars.

  1. Add the standards used in each part of the research method.

Response: Measurement standards are indicated in the text.

  1. The standard deviation should be added to the results.

Response: Thanks a lot. Experimental measurement curves have measurement errors.

  1. Table 1 should be deleted. Printing parameters are summarized in a table.

Response: Table 1 was removed.

  1. The existence of figures 2 and 3 is not required. Be omitted.

Response: We agree, the Figs. 2-3 were omitted.

  1. Section 2.5 should also be summarized. The equations in this section should be deleted. In this section, cooling rate, heating rate, load mode, frequency, temperature range, strain rate and ASTM should be presented.

The next section should be summarized.

Response: It was done. We tried following to Reviewer recommendations.

  1. Figures 4 and 5 should be merged.

Response: We merged all samples views before and after mechanical testing.

  1. Figures 6-8 should be deleted.

Response: It was done.

  1. Figures 10-11 should be deleted.

Response: We merged the Figs. 10-11. In connection with the deletion of previous Figures and after enumeration, this Figs. 3-4 are.

  1. Figures 12 and 13 for two variable printing parameters are meaningless. The results should only be presented in the table.

Response: Figs. 12-13 were moved in Appendix A. We summarized ANOVA results in Tables 5-6 (new enumeration).

  1. The conclusion is also long and should be corrected like the abstract.

Response: Conclusion section was concretized.

  1. How has the used model been verified? How are the structural equation and its parameters selected?

Response Numerical model and defining equations for DMA simulation was chosen based on approaches of other research studies [34, 35, 37].

Mooney-Rivlin model for simulation of uniaxial tensile tests in Ansys was selected based on comparing convergence for 3, 5, 9 parameters models (Prony series, Fig. 6). The 3-parametrs Mooney-Rivlin model showed greater deviation from experimental curves (Fig. 2) compare to 5-parameters model. 7-parameters model gave particularly same values as 5-parameters model, but with a bigger computational resource.

  1. The research method section is long and should be summarized.

Response: After fulfilling the previous recommendations of the Reviewer, the section was significantly reduced.

We are sincerely grateful to the honorable Reviewer #2 for their careful and deep viewing of the paper, helpful comments and constructive remarks. We do believe that they really helped us to improve the paper and be more persuasive in reporting our results for the readers. We highly appreciate your time and efforts spent for editing the manuscript.

Reviewer 3 Report

Dear authors,

The paper is interesting and I propose it for publication after addressing the highlighted comments:

1. Introduction: I propose to add more references in the literature review as it is not sufficient for an introduction. Please also include the following reference regarding the mechanical behavior of the 3D-printed parts:

https://doi.org/10.3390/ma15248722

2. How you have chosen the selected values for printing the specimen ? it should be mentioned in the paper.

3. The characterization techniques as well as FEM are well explained.

4. Figure 9: I propose to separate it into two parts as it is not clear enough.

5. Figure 14: I propose to revise the legend of the curves as the values and related info are not clear enough.

6. Results and discussions are well explained and structured.

Author Response

Reviewer 3

Dear authors,

The paper is interesting and I propose it for publication after addressing the highlighted comments.

Response: Thank you for your comments and recommendations, we appreciate your involvement.

  1. Introduction: I propose to add more references in the literature review as it is not sufficient for an introduction. Please also include the following reference regarding the mechanical behavior of the 3D-printed parts:

https://doi.org/10.3390/ma15248722

Response: According to recommendations of respected Reviewers #1 & #3 the Introduction section was rewrote.

  1. How you have chosen the selected values for printing the specimen? it should be mentioned in the paper.

Response: Thanks a lot for right question. Printing parameters were chosen based on the preliminary literary analysis and own experience in working with a 3D printer. The literature states that the most important among all the printing parameters are layer thickness, infill, temperature [28], [29]. The values for the infill density were selected based on preliminary attempts to evaluate the quality of printed parts. So details with a density of less than 20% turned out to be unacceptable, and huge difference in strength was not observed between 80% and 100%, only time consumption and waste material were highlighted.

  1. The characterization techniques as well as FEM are well explained.

Response: Thanks for your friendly comments.

  1. Figure 9: I propose to separate it into two parts as it is not clear enough.

Response: It was corrected.  For better understanding, we have added a title to each part of the plot. TMA and DSC analyses were presented together to show the different nature of processes with temperature raise.

  1. Figure 14: I propose to revise the legend of the curves as the values and related info are not clear enough.

Response: It was corrected.  Here we have additionally added the title of curves.

  1. Results and discussions are well explained and structured.

Response: Thanks for your friendly comments.

We are sincerely grateful to the honorable Reviewer #3 for their careful and deep viewing of the paper, helpful comments and constructive remarks. We do believe that they really helped us to improve the paper and be more persuasive in reporting our results for the readers. We highly appreciate your time and efforts spent for editing the manuscript.

Round 2

Reviewer 1 Report

The work has been modified according to the observations and in its current form it can be published.

Reviewer 2 Report

Accept.